# Dual Effect of Taxifolin on ZEB2 Cancer Signaling in HepG2 Cells

**DOI:** 10.3390/molecules26051476

**Published:** 2021-03-09

**Authors:** Zdenek Dostal, Martin Sebera, Josef Srovnal, Katerina Staffova, Martin Modriansky

**Affiliations:** 1Department of Medical Chemistry and Biochemistry, Faculty of Medicine and Dentistry, Palacký University, 77515 Olomouc, Czech Republic; zjedna@post.cz; 2Faculty of Sport Studies, Masaryk University, 60177 Brno, Czech Republic; sebera@fsps.muni.cz; 3Institute of Molecular and Translational Medicine, Faculty of Medicine and Dentistry, Palacký University, 77515 Olomouc, Czech Republic; josef.srovnal@upol.cz (J.S.); katerina.staffova@upol.cz (K.S.)

**Keywords:** polyphenols, Affymetrix GeneChip™ miRNA 3.0 Array, Hep G2 cells, primary cultures of human hepatocytes, ZEB2

## Abstract

Polyphenols, secondary metabolites of plants, exhibit different anti-cancer and cytoprotective properties such as anti-radical, anti-angiogenic, anti-inflammation, or cardioprotective. Some of these activities could be linked to modulation of miRNAs expression. MiRNAs play an important role in posttranscriptional regulation of their target genes that could be important within cell signalling or preservation of cell homeostasis, e.g., cell survival/apoptosis. We evaluated the influence of a non-toxic concentration of taxifolin and quercetin on the expression of majority human miRNAs via Affymetrix GeneChip™ miRNA 3.0 Array. For the evaluation we used two cell models corresponding to liver tissue, Hep G2 and primary human hepatocytes. The array analysis identified four miRNAs, miR-153, miR-204, miR-211, and miR-377-3p, with reduced expression after taxifolin treatment. All of these miRNAs are linked to modulation of ZEB2 expression in various models. Indeed, ZEB2 protein displayed upregulation after taxifolin treatment in a dose dependent manner. However, the modulation did not lead to epithelial mesenchymal transition. Our data show that taxifolin inhibits Akt phosphorylation, thereby diminishing ZEB2 signalling that could trigger carcinogenesis. We conclude that biological activity of taxifolin may have ambiguous or even contradictory outcomes because of non-specific effect on the cell.

## 1. Introduction

Polyphenols are generally secondary metabolites of plants where they play different roles such as in protection against oxidative stress, infection or UV-light [1]. These natural compounds are abundant in fruits, vegetables, plant derived beverages, etc., and thus form an important part of the human diet. Average consumption of polyphenols can exceed 1 g per day [2,3]. Their positive effects were demonstrated by many in vitro and in vivo experiments [4,5]. The most often discussed effect is antioxidant activity [6]. Other interesting effects include anti-microbial, anti-inflammatory, anti-cancer, or cardioprotective properties [6,7,8,9]. Some authors even claim that the basis of biological effects of polyphenols is modulation of cell signalling rather than anti-oxidant properties [10]. During the past several years many studies have described the ability of polyphenols and other natural compounds to modulate microRNA (miRNA) expression [10,11,12,13].

MiRNA was first described by Lee et al. [14]. While working with *C. elegans* the authors discovered an unusual RNA molecule encoded by gene *lin-4* that has the ability to modify expression of *lin-14* gene product. Gradually 1982 miRNA precursors were described (according to miRBase 22 [15]). Interestingly, the effects of miRNA are quite a complex mosaic because one miRNA molecule is capable of modulating the expression of many protein targets, while several molecules of miRNA can modulate the same protein target [16,17]. Recent articles discuss a high proportion of miRNAs as tumor suppressors or oncogenes depending on their rate of expression and function of their target proteins [18]. Sometimes a miRNA can be designated as tumor suppressor and oncogene at the same time within different cell types. In other words, the role of these specimens/molecules could be apparently conflicting as is the case of the miR-29b family [19,20].

ZEB2 protein, also known as Zinc finger E-box-binding homeobox 2 or SIP1, is a transcription factor [21,22]. It plays an important role in epithelial/mesenchymal transition (EMT). EMT is a complicated process regulated by several transcription factors such as ZEB, Snail or Twist, and epigenetics [23]. This phenomenon appears usually in later stages of cancer development and is often linked to more aggressive malignancies with high migration potential [24]. The transition is characterized by divergent expression of multiple integral proteins, such as vimentin upregulation or E-cadherin downregulation [22,25]. On the other hand, EMT is also part of the normal development of the organism, especially during embryonic development or wound healing [26]. Moreover, there are many miRNAs that directly regulate ZEB2 expression, miR-30a, miR-141, and miR-335 to name a few [27,28,29].

miRNAs are important molecules in modulation of protein expression and could be affected by polyphenols. Hence, we decided to test the possible effect of taxifolin and quercetin, two fairly abundant polyphenols of the human diet, in hepatocyte models. Quercetin was chosen as a major representative of the flavonoid group and taxifolin is its reduced derivative. We hypothesized that miRNA expression altered by polyphenols may be beneficial for different types of cells and tissues or result in modified behaviour, which may correspond with protective properties.

## 2. Results

### 2.1. Quercetin and Taxifolin Are Not Toxic in Hep G2 Cells

We evaluated the effect of quercetin and taxifolin on viability of Hep G2 by performing MTT (3-(4,5-dimethylthiazol-2-yl)-2,5-diphenyltetrazolium bromide) and neutral red assays over a range of concentrations up to 100 µM for both compounds. Our results show that quercetin caused toxicity at doses higher than 40 µM (Figure 1A). On the other hand, taxifolin was considered non-toxic substance in all concentrations tested (Figure 1B). Hence, we can conclude that physiologically attainable concentration of tested polyphenols, i.e., 1 µM, is not harmful.

### 2.2. Analysis of miRNAs Expression Profile in Affymetrix GeneChip™ miRNA 3.0 Arrays

In the next step of the study, we used Affymetrix GeneChip™ miRNA 3.0 Arrays for miRNA expression profile analysis after treatment with 1 µM concentration of quercetin and taxifolin. The chosen microarray platform contains specific probes for more than 1700 human miRNAs. The data processing consisted of normalization of each miRNA to negative control followed by comparison of distinctly modulated miRNAs with their validated targets. There are 30 miRNAs with ZEB2 as validated target by using the miRTarBase [30] and based on other recent literature (Table 1 and Appendix A, Table A1). The selected miRNAs met the following criterion—effect on ZEB2 expression was validated by reporter assay, Western blot or qPCR technique. Our gene chip analysis revealed twelve miRNAs of the 30 with ZEB2 as validated target, deregulated more than 1.5 times by taxifolin compared to control. Four of them were downregulated and seven were upregulated in Hep G2 cell model. In primary hepatocytes one and two miRNAs were up- and downregulated, respectively. Overall, hsa-miR-211 and hsa-miR-377 were downregulated in both models (Table 1).

### 2.3. Taxifolin Modulates ZEB2 Expression in Hep G2 Cells But Not in Primary Cultures of Human Hepatocytes

Our results from the gene chips suggested modulation of ZEB2 protein expression via miRNA. Therefore, we evaluated the impact of taxifolin and quercetin on ZEB2 protein expression. We performed 24 h incubation with tested compounds followed by whole lysates preparation and Western blot analysis. In the Hep G2 cells taxifolin caused dose dependent upregulation of ZEB2 expression (Figure 2), but quercetin did not (Appendix A, Figure A1). There were no significant effects by either tested compound on ZEB2 expression in primary cultures of human hepatocytes (data not shown).

### 2.4. ZEB2 Upregulation Did Not Cause the Epithelial to Mesenchymal Transition

ZEB2 protein is important in EMT [36], a process that occurs in advanced stages of cancer development. It is linked to higher aggressiveness of the malignancy and mobility of cancer cells [37]. Therefore, we tested for these properties in experiments with xCELLigence system, wound healing assay and Western blot evaluation of vimentin protein expression as markers of EMT. Surprisingly, Western blot results showed dose dependent decrease of vimentin expression (Figure 3). That suggests the EMT process does not occur in treated Hep G2 cells and another, perhaps stronger, signalling pathway exists that regulates vimentin expression. In addition, wound healing assay after 24 h, as well as cell morphology after 24 h treatment showed only negligible changes (Appendix A, Figure A2B,C). On the other hand, xCELLigence results after 24 h incubation show that taxifolin dose dependently, albeit very slightly, enhances growth of Hep G2 cells in the concentration range from 1 to 10 µM. The data, however, failed to reach statistical significance when compared to negative control with dimethyl sulfoxide (DMSO) (p_10 µM_ = 0.065, Appendix A, Figure A2A).

### 2.5. Taxifolin Negatively Regulates Vimentin Expression via Akt Dephosphorylation

The effect of taxifolin treatment on the modulation of vimentin expression, a downstream target of ZEB2 [38], was contrary to our preceding results. A search of the literature revealed several signalling proteins, transcription factors or miRNAs, which modulate vimentin expression while being regulated by the taxifolin treatment. These are miR-375, c-jun, NFκB, β-catenin and dephosphorylation of Akt. We endeavoured to test which of these could be responsible for the dual effect of taxifolin. Our results show that Akt dephosphorylation is the decisive signal for vimentin downregulation (Figure 4). Expression of another potential candidate, wild-type β-catenin, was downregulated in the cytosol fraction and a similar trend was observed in the nuclear fraction (Appendix A, Figure A3A,B). Further, the data for wild-type β-catenin suggest a slight, statistically insignificant relocation of the protein with taxifolin. On the other hand, expression of the truncated form of β-catenin was unchanged (Appendix A, Figure A3C,D). c-jun (Figure 4A), NFκB, and miR-375 showed non-significant differences (data not shown). Moreover, experiments with Akt inhibitor GSK690693 showed correlation between Akt activity and vimentin expression that is consistent with our hypothesis (Appendix A, Figure A5A). Activity of Akt kinase was evaluated via phosphorylation of CREB (Ser133) as a downstream target (Appendix A, Figure A6) [39].

### 2.6. Reduction of ZEB2 Expression by miR-377 Precursors Is Restored by Taxifolin in a Dose-Dependent Manner

All experiments performed so far have not shown an association between the miRNA, expression of ZEB2 and taxifolin. Therefore, we transiently transfected Hep G2 cells with two different miRNA precursors or negative control. The transfected cells were incubated with selected concentrations of taxifolin. The cells transfected with miR-377 precursors showed the expected reduction in ZEB2 expression that was restored by taxifolin in a dose dependent manner. On the other hand, miR-211 transfected cells showed neither ZEB2 down-regulation, nor taxifolin restoration of the effect. Induction of ZEB2 by taxifolin was present after transfection of the cells with negative control precursors as well.

## 3. Discussion

Experimental design invokes a crucial decision on what concentration or concentration range to use. Preferably one that has been used by predecessors or one that was established as attainable under physiological situation in the serum. Literature search indicated that such a concentration for quercetin and taxifolin is approximately 1 µM [40,41,42]. This was the main reason why we chose 1 µM concentration for our experiments performed with gene chip technology. The initial toxicity experiment was designed to ensure that any toxic effect of the tested compounds appears at concentrations at least 20-fold higher. Another crucial decision was the selection of in vitro model. Because liver is the organ deemed responsible for majority of xenobiotic metabolism, we chose cell models that represent liver. Moreover, the liver is the first organ exposed to effects of these compounds following intestinal absorption. Hence, there is high probability that the tissue is exposed to the highest amount/concentration of the compounds, most likely including the concentration range used in our experiments.

miRNA array analysis is a comprehensive tool for extensive analysis of miRNAs expression. It offers a huge set of probes for different miRNAs fixed to a glass matrix in precisely defined positions from which the amount of single miRNAs can be determined based on fluorescence intensity. Our data revealed several miRNAs that are linked to ZEB2 expression as deregulated (Table 1 and Table A1). Hence, every miRNA can play a different role in the ZEB2 expression in a particular tissue. The final result is dependent on a “final vector” composed from all miRNA effects. Therefore, the apparent effect of a treatment on a single miRNA or even a set of miRNAs must be verified through independent approaches as it could be diminished or abolished altogether because of other influences.

In our case the Western blot analysis demonstrated a dose dependent induction of ZEB2 expression during taxifolin treatment giving support to the working hypothesis of modulating EMT via miRNA (Figure 2). Quercetin, despite the same trend of influence on ZEB2 expression, did not reach statistical significance because of high variability (data not shown) which may be related to the different stability of quercetin and taxifolin in the culture medium [43]. ZEB2 is an important part of EMT signalling and plays a significant role in carcinogenesis. The EMT is characterized by different expression of several integral proteins such as downregulation of E-cadherin and upregulation of N-cadherin that serve as markers of this important cell process. The result of EMT is loss of epithelial and gain of mesenchymal phenotype. Epithelial dedifferentiation is associated with the regulation of various intercellular junction components that results in facilitating cell migration [36]. Another EMT marker is vimentin, known as an important part of the cytoskeleton and a player in wound healing or metastasizing of cancer cells [44]. According to our hypothesis, vimentin upregulation should correspond with expression of ZEB2 [38]. However, vimentin expression did not follow our expectations indicating the existence of another, perhaps stronger, mechanism or signalling pathway, which can reverse ZEB2-EMT potential (Figure 3).

The pathways that affect vimentin include NF-κB, β-catenin, c-Jun, and Akt signalling, all of which were shown to be modulated by taxifolin, see below. NF-κB plays a relatively important role in cancer, but mutations causing direct activation of NF-κB are rare in solid tumors compared to blood malignancies [45]. In contrast, several publications suggested that consecutive NF-κB activity exists in Hep G2 cells. For example, incubation of this cell line with inhibitor IKK-2 reduces NF-κB activity [46]. In our experiments, the effect of taxifolin on NF-κB was negative as we expected.

Several publications reported negative effect of taxifolin on β-catenin expression in breast carcinoma cell lines or colorectal carcinoma cells HCT116 and HT29 [47,48]. β-catenin plays an important role in the regulation of many tumor-related events and its activation may result in upregulation of vimentin expression [47]. Generally, β-catenin activity is regulated by several different mechanisms. The first mechanism is rate of β-catenin expression; the second, E-cadherin sequesters nascent β-catenin in the cytosol; and the third possibility is degradation of the protein mediated by the destruction complex that is negatively associated with the Wnt signalling pathway. The inhibition of the last two mechanisms is followed by redistribution of β-catenin between cytosol and nucleus [47,49] thereby affecting its function as transcription factor and co-activator for the TCF/Lef [49]. Lastly, Wnt/β-catenin pathway was shown to be linked to PI3K/Akt pathway with glycogen synthase kinase 3 (GSK-3) as downstream and effector molecule [50]. Indeed, lower Akt activity was accompanied by β-catenin degradation (Figure A3A,B). However, the Hep G2 cell line is specific for its truncated form, which is insensitive to GSK-3 (Akt signalling, Appendix A, Figure A3C,D), [51]. Hence, the contribution of β-catenin appears to be limited due to reduced expression of wild-type β-catenin and predominance of its truncated form.

c-Jun is subjected to Akt-dependent regulation as well [52]. However, the protein’s expression was not significantly changed in our experiments (Figure 4). It seems that these signalling nodes of the Akt network are modulated in a context-dependent manner as discussed in several publications [50,53].

Finally, as a serine/threonine kinase, Akt alone mediates many different effects including cardioprotection, carcinogenesis, regulation of metabolism, expression of ABC transporters or vimentin via more than one hundred downstream effector molecules [54,55,56,57]. According to the literature, taxifolin can mediate Akt dephosphorylation that directly contributes to depletion of vimentin [57,58] as confirmed by experiments with the Akt inhibitor GSK690693 (Appendix A, Figure A4). The involvement of Akt in the regulation of EMT is much more complex than just the modulation of vimentin expression. Reduced activity of Akt is accompanied by upregulation of E-cadherin and downregulation of N-cadherin resulting in mesenchymal to epithelial transition [59].

Last set of experiments focused on effect of selected miRNA precursors in the context of taxifolin-treated cells. The aim was to verify whether these miRNAs are involved in taxifolin-induced upregulation of ZEB2. We chose miR-377 and miR-211 for these experiments because the data from the miRNA arrays showed their reduced expression for both models. The results for cells transfected with miR-377 precursors confirmed our hypothesis of the involvement of miR-377 (Figure 5). Surprisingly, and contrary to previously published data, we failed to demonstrate an association between transfection of miR-211 precursors, taxifolin, and ZEB2 expression [34]. Finally, Hep G2 cells transfected by negative control precursors showed same dose dependent effect of taxifolin as non-transfected cells.

Biological activity of polyphenols is in part associated with modulation of miRNA expression. The overall effect is, however, context-dependent. Data presented in this work suggest that upregulation of ZEB2 protein by taxifolin treatment, but not by quercetin, probably proceeds via miR-377 in Hep G2 cell model. On the other hand, primary human hepatocytes showed only non-significant changes in ZEB2 expression even though two of the miRNAs, which are linked to ZEB2, were downregulated. This could be explained by tremendous heterogeneity of primary cultures derived from tissue that underwent unknown sets of prolonged treatments (lifestyle, diet, pharmacotherapy, etc.).

The overall result in more homogenous, in terms of handling and treatments, cell line does not support the initial finding of modulation of ZEB2 expression via miRNA. This is inferred from the lack of EMT one would expect to follow the ZEB2 modulation. We show that the effect of ZEB2 is reversed due to the second effect of taxifolin: inhibition of Akt signalling. Which, in turn, suggests that Akt signalling has stronger effect on EMT than ZEB2 transcription factor alone. It points out that a singular effect of, e.g., taxifolin on subcellular level may have no positive or negative outcome on cellular or tissue level because of pleiotropic effects. In our case the overall effect of taxifolin on EMT is neither negative nor positive, for the substance affects two apparently counteracting signals.

## 4. Materials and Methods

### 4.1. Chemicals

Quercetin, taxifolin, dimethyl sulfoxide (DMSO), Dulbecco’s Modified Eagle Medium—high glucose, Nutrient Mixture F-12 Ham, William’s E Medium, penicillin (10,000 units/mL) streptomycin (10 mg/mL) solution, bovine serum albumin, ethanol, isopropanol, TRI Reagent, chloroform, nuclease free water, collagen, methanol, glucose, glutamine, sodium pyruvate, dexamethasone, holo-transferrin, ethanolamine, insulin, glucagon, ascorbic acid, linoleic acid, amphotericin B, fluconazole, ammonia, ammonium persulfate, acrylamide/bis-acrylamide (37.5:0.9), 3-(4,5-dimethylthiazol-2-yl)-2,5-diphenyltetrazolium bromide (MTT), and neutral red were from Merck (Darmstadt, Germany). Non-essential amino acids, Pre-miR™ miRNA Precursor—hsa-miR-377-3p, Pre-miR™ miRNA Precursor—hsa-miR-211-5p, Pre-miR™ miRNA Precursor Negative Control #1, Lipofectamine 2000, TaqMan™ MicroRNA Assay for hsa-miR-375, TaqMan™ MicroRNA reverse transcription kit and TaqMan™ Universal PCR Master Mix, no AmpErase™ UNG were obtained from Life-Technologies (Prague, Czech Republic). Fetal bovine serum was purchased from Bio-Tech (Prague, Czech Republic).

### 4.2. Cell Culture Conditions

Hep G2 cells were obtained from ECACC (No. 85011430 Human Caucasian hepatoblastoma cell line) via Sigma-Aldrich (St. Louis, MO, USA) as provider. Dulbecco’s Modified Eagle Medium—high glucose containing supplements was used for cultivation of the cells (supplements: 10% fetal bovine serum, 5% non-essential amino acids, 100 units/mL of penicillin, and 0.1 mg/mL of streptomycin). Cells were incubated in humidified incubator with 5% CO_2_ atmosphere at 37 °C and passaged every 2 to 4 days.

### 4.3. Primary Cultures of Human Hepatocytes

Isolation of primary human hepatocytes was performed according to a two-step collagenase perfusion followed by hepatocyte release and centrifugation cleaning steps. For culturing, collagen-coated culture dishes were used, and 2 × 10^5^ cells were seeded per every cm^2^. The cells were plated in ISOM medium containing dexamethasone and insulin supplemented with 10% fetal bovine serum, 100 units/mL of penicillin, and 0.1 mg/mL of streptomycin. The serum containing medium was replaced after 24 h stabilization by serum-free medium with composition described above. Further, cells were exposed to tested compounds or vehicle control (DMSO).

Primary human hepatocytes were obtained from multiorgan donors in accordance with the permission from ethics committee of University Hospital Olomouc (reference number 119/07).

### 4.4. Cell Viability Assays

MTT assay is based on reduction of MTT (5 mg/mL) to formazan by mitochondrial oxidoreductases. The reduction takes place only in live and metabolically active cells [60]. Incubation with polyphenols was followed by PBS washing step and addition of freshly prepared solution of MTT in serum-free medium solution in ratio of 1:10. The mixture was pipetted in to each well, treated with tested compounds or DMSO or Triton-X100 solution, and cells were incubated for 2 h at 37 °C. The mixture was aspirated, and formazan crystals were dissolved in DMSO/0.1% NH_3_ solution. Absorbance was measured at 540 nm.

Neutral red assay is pursuant to accumulation of neutral red dye within the cells. Incubation with polyphenols was followed by PBS washing step and addition of freshly prepared neutral red solution:serum-free medium in 3:8 ratio. The final solution was pipetted into each well, treated with tested compounds or DMSO or Titon-x100 solution, and cells were incubated for 2 h at 37 °C. The solution was discarded, and cells were washed with mixture of 0.5% formaldehyde +1% CaCl_2_ solution (200 µL, 1:1 ratio) and then dissolved in 50% methanol containing 1% CH_3_COOH. Absorption was measured at 550 nm.

### 4.5. xCELLigence System

xCELLigence Real-Time Cell Analysis Instrument (Accela, Prague, Czech Republic) utilizes E-plates with well bottoms containing gold electrodes for measuring of impedance. The recorded data are subsequently recalculated to a cell index and plotted versus time. The arbitrary unit of “cell index” is proportional to the number of cells covering the electrodes of E-Plates. Every “E-Plate view 16 PET” contains sixteen wells identical in size to those in 96 well plates. Each trace of cell index versus time was subjected to evaluation in RTCA software 1.2.1. The results are expressed as doubling time in selected time period normalized to negative control (see Figure 6). The taxifolin addition is accompanied by exchange of media which results in formation of an artefact in the traces. Hence, the doubling point analysis was started when the cell index of all traces was stabilized and growing. The very last point of tested data set was 24 h later. In this case, the data represent proliferation rate of treated cells.

### 4.6. Bright-Field Microscopy

Zeiss AxiovertC microscope (2.5×, 5×, and 40× objectives) with a Zeiss AxioCam ICM1 (Zeiss GmbH, Jena, Germany) was used for evaluation of cell morphology changes and wound healing assays.

### 4.7. RNA Isolation

The TRI-Reagent was utilized for RNA isolation and purification, according to manufacturers recommended protocol. The established standard method is based on phenol-chloroform extraction published by Chomczynski et al. [61]. Purity and amount of RNA was measured by Implen Nanophotometer (Implen GmbH, München, Germany). RNA used for experiments had purity higher than 1.8 (A260/A280).

### 4.8. miRNA Array Analysis

miRNA array analysis was done with Affymetrix GeneChip™ miRNA 3.0 arrays. The method used oligonucleotide probes attached to glass matrix, organized into a specific pattern. At the beginning of the experiment, each oligonucleotide in the sample was enriched with the polyA tail that was connected with modified biotin containing polyT. This step was followed by hybridization of sample miRNAs with probes on the array. Quantity of different miRNA was detected via fluorescently labelled streptavidin. Detection was done in Affymetrix scanning machine GeneChip™ Scanner 3000 7G (Affymetrix [part of ThermoFisher Scientific company], Santa Clara, CA, USA). We used manufacturer’s recommended protocol for sample preparation, hybridization, staining and detecting of our oligonucleotides.

### 4.9. RT-PCR

Reverse transcription is a standard molecular biology procedure that was performed for cDNA synthesis from miRNA template. The specificity was ensured by TaqMan™ MicroRNA Assay primers. The product was then used for real-time PCR analysis with specific TaqMan™ primers and probes. For actual RT-PCR reaction Roche Light Cycler 480 (Roche, Basel, Switzerland) was used. The delta-delta Ct method was used for quantification [62].

### 4.10. Immunodetection

Whole cell lysates or subcellular extracts were used for immunodetection. Sample separation was done at 10% or 12.5% SDS-PAGE gels and then transferred onto PVDF membranes. Individual proteins ZEB2, NFκB, c-jun, vimentin, Akt, p-Akt (Ser473), p-CREB (Ser133), β-catenin, β-tubulin, GAPDH, and actin were detected using the corresponding primary antibody at 1:1000 and secondary antibody at 1:10,000 dilution. Primary and secondary (HRP-linked) antibodies were obtained from Santa Cruz biotechnologies (Dallas, TX, USA) or Cell Signaling Technology (Danvers, MA, USA). Detection was performed using commercially available substrate solution ImmunoCruz (Santa-Cruz biotechnology) and Cerastream Kodak BioMax light films (Sigma-Aldrich, St. Louis, MO, USA) or Fuji medical X-ray films (FujiFilm, Tokyo, Japan).

### 4.11. Transfection

Precursors of hsa-miR-211, hsa-miR-377, or negative control were transfected into the cells using Lipofectamine 2000. Transfection of cells in suspension was performed according to the manufacturer’s protocol. Incubation with transfection mixture lasted 6 h. More detailed protocol was described elsewhere [63].

### 4.12. Statistical Analysis

One way ANOVA with Tukey´s post hoc test for statistic evaluation of our datasets was done in the Statistica 12 software (StatSoft CR s.r.o. [part of Dell company], Prague, Czech Republic).

## Figures and Tables

**Figure 1 molecules-26-01476-f001:**
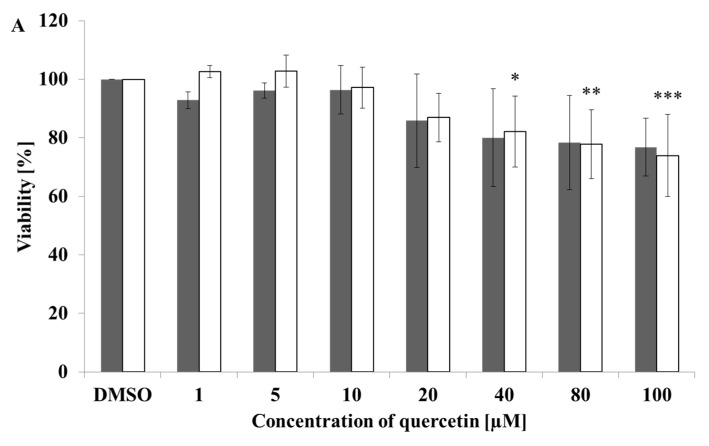
Cytotoxicity of tested compounds on Hep G2 cells evaluated with MTT and Neutral red assays. Panel (**A**) represents cytotoxicity of quercetin and panel (**B**) cytotoxicity of taxifolin evaluated with MTT/Neutral red assay that are shown as filled bars/empty bars, respectively. Each bar represents mean ± SD of three independent experiments. Cells were incubated with tested compounds for 24 h; in each experiment triplicate measurements were performed. * *p* < 0.05(** *p* < 0.01/*** *p* < 0.001) versus negative control.

**Figure 2 molecules-26-01476-f002:**
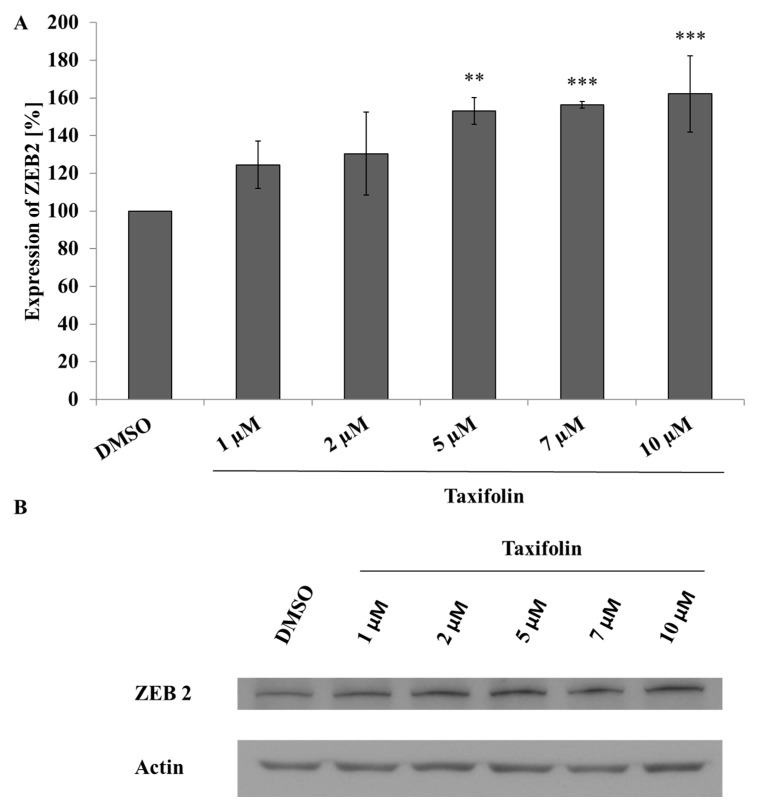
Modulation of ZEB2 expression by taxifolin in Hep G2 cells. Panel (**A**): Hep G2 cells were seeded and incubated with different concentrations of taxifolin. Negative control contains only dimethyl sulfoxide (DMSO). ZEB2 expression was evaluated by Western blot after 24 h of incubation. Each bar represents mean ±SD of three independent experiments. Panel (**B**): representative Western blot. ** *p* < 0.01(*** *p* < 0.001) versus negative control.

**Figure 3 molecules-26-01476-f003:**
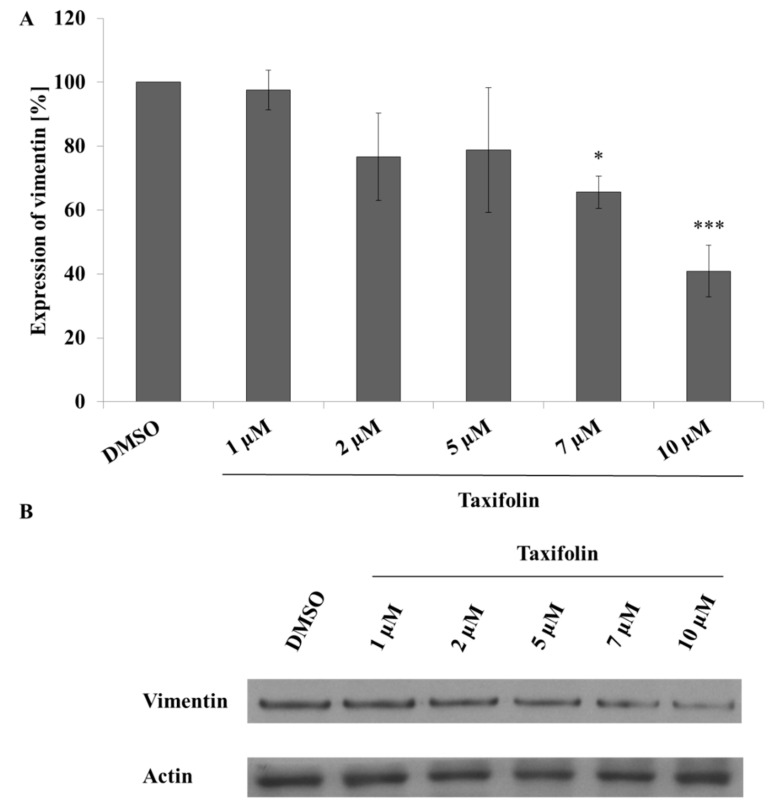
Modulation of vimentin expression by taxifolin. Panel (**A**): Hep G2 cells were seeded and incubated with different concentrations of taxifolin or negative control (contains only DMSO). Vimentin expression was evaluated by Western blot after 24 h of incubation. Each bar represents mean ± SD of three independent experiments. Panel (**B**): representative Western blot. * *p* < 0.05 (*** *p* < 0.001) versus negative control.

**Figure 4 molecules-26-01476-f004:**
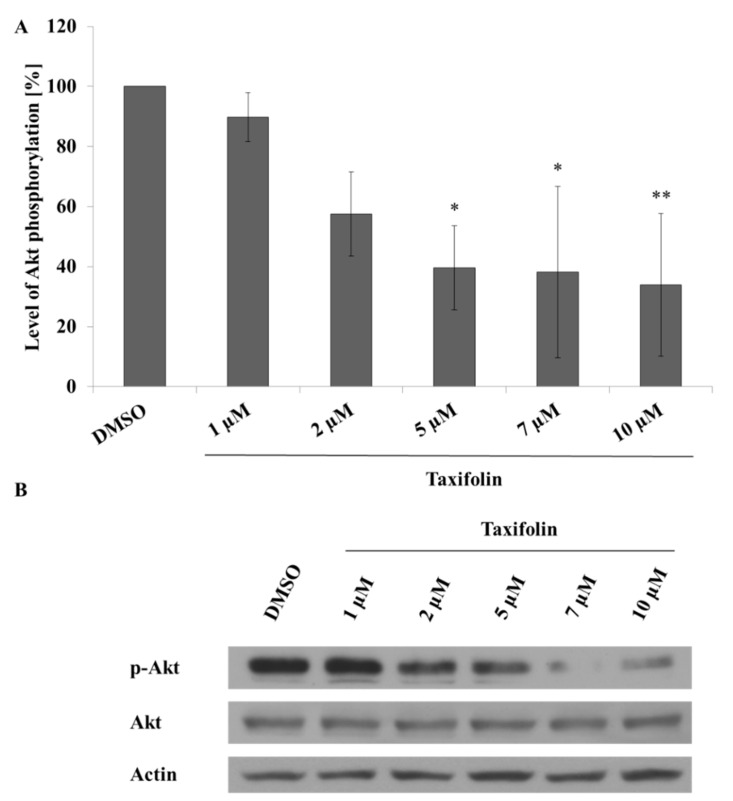
Modulation of Akt phosphorylation (Ser473) by taxifolin. Panel (**A**): Hep G2 cells were seeded and incubated with different concentrations of taxifolin or negative control (contains only DMSO). Akt phosphorylation was evaluated by Western blot after 24 h of incubation. Each bar represents mean ± SD of three independent experiments. Panel (**B**): representative Western blot. * *p* < 0.05 (** *p* < 0.01) versus negative control.

**Figure 5 molecules-26-01476-f005:**
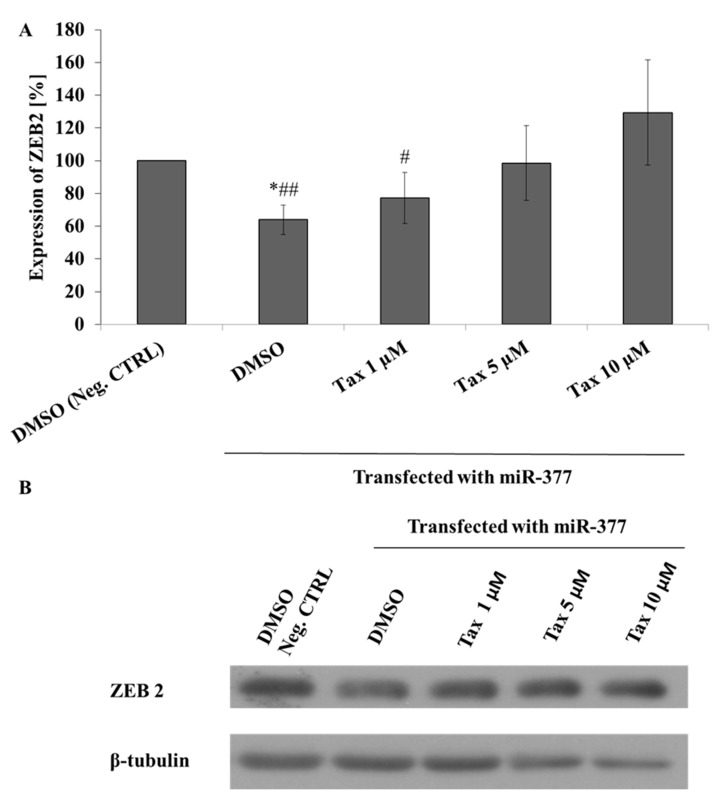
Modulation of ZEB2 in miR-377 transfected cells by taxifolin. Panel (**A**): Hep G2 cells were transfected with miR-377 precursors or negative control and incubated with different concentrations of taxifolin. ZEB2 expression was evaluated by Western blot after 24 h of incubation. Each bar represents mean ± SD of three independent experiments. Panel (**B**): representative Western blot. * *p* < 0.05 versus negative control and ^#^
*p* < 0.05 (^##^
*p* < 0.01) versus cells transfected with miR-377 precursors and incubated with 10 µM taxifolin.

**Figure 6 molecules-26-01476-f006:**
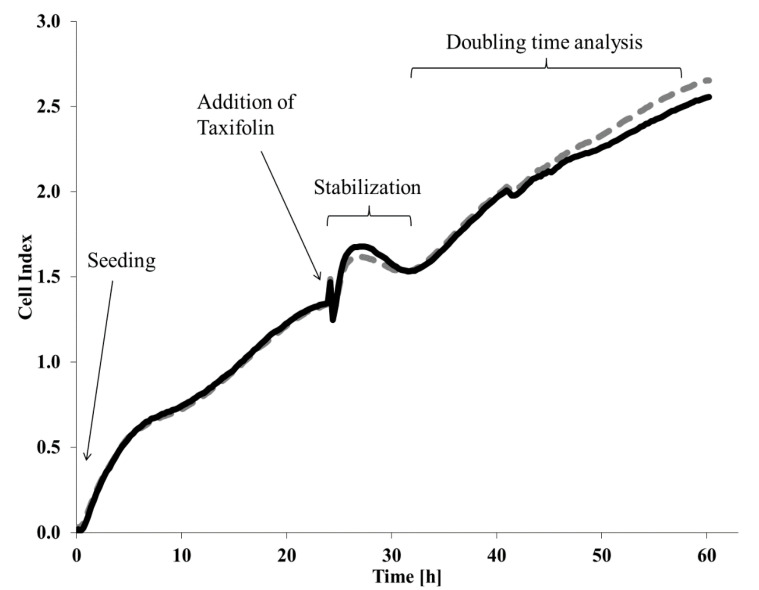
Example of xCELLigence trace recording. The figure contains xCELLigence traces (curves) to illustrate the evaluation method. The dashed line represents treatment of 10 μM taxifolin and the solid line represents DMSO treated control.

**Table 1 molecules-26-01476-t001:** Expression analysis of miRNAs that contain ZEB2 as their validated target in Hep G2 and human hepatocytes.

miRNA	Mean Relative Expression Compared to Control								References
	Quercetin				Taxifolin				
	Hep G2		Human Hepatocytes		Hep G2		Human Hepatocytes		
	Mean	SD	Mean	SD	Mean	SD	Mean	SD	
hsa-miR-129-5p	**2.42**	2.11	0.94	0.27	**4.53**	6.58	1.07	0.17	[31]
hsa-miR-139-5p	1.40	0.16	1.35	0.86	**2.14**	1.06	1.34	1.09	[32]
hsa-miR-141-3p	**1.90**	0.96	0.98	0.17	**1.56**	1.21	1.00	0.42	[30]
hsa-miR-153	1.02	0.37	0.72	0.24	0.67	0.27	1.39	0.89	[30]
hsa-miR-154-5p	1.40	0.44	**1.73**	1.09	**1.63**	0.48	1.45	0.60	[30]
hsa-miR-200c-3p	1.00	0.00	**1.59**	0.47	1.00	0.04	**1.85**	0.70	[30]
hsa-miR-204	0.79	0.55	0.78	0.03	0.68	0.32	0.86	0.38	[33]
hsa-miR-211	1.17	0.72	0.91	0.42	0.59	0.20	0.74	0.33	[34]
hsa-miR-335-5p	1.47	1.49	0.77	0.20	**2.10**	2.98	0.84	0.14	[30]
hsa-miR-377-3p	1.42	1.16	0.65	0.19	0.64	0.43	0.57	0.05	[35]
hsa-miR-590-3p	1.16	0.54	1.15	0.31	**2.09**	0.81	0.95	0.08	[30]
hsa-miR-4782-3p	1.47	0.55	0.75	0.21	**1.73**	0.96	1.06	0.37	[30]

The values represent mean of three independent samples/arrays compared to control. Expression changes lower than 0.75 are highlighted in grey boxes, expression changes higher than 1.5 are highlighted in bold. Table includes differences against current nomenclature because we used Affymetrix GeneChip™ miRNA 3.0 Arrays which do not contain 3p/5p specification of hsa-miR-153, hsa-miR-211, hsa-miR-215, and any specification for hsa-miR-203. The references listed for each row refer to the sources where the ZEB2 protein has been validated as a target for a given miRNA.

## Data Availability

All the relevant data are contained in the text, including Appendix A. Raw data will be provided by the corresponding author upon request.

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
