# Peer review of "Dual Effect of Taxifolin on ZEB2 Cancer Signaling in HepG2 Cells"

_molecules, 2021, doi:10.3390/molecules26051476_

Round 1

Reviewer 1 Report

The authors present a study exploring the effect of polyphenols (specifically taxifolin) on ZEB2 signalling and its association with miRNA expression. While the authors begin by assessing both taxifolin and quercetin, they pursue the study of taxifolin due to its ability to alter the expression of ZEB2. While quercetin did not change ZEB2 expression, the authors do not demonstrate any evidence to support this (I would recommend a western blot of ZEB2 expression with quercetin to drive home the message that only taxifolin alters ZEB2 expression) nor do they discuss a possible explanation as to why the two compounds may impose differential effects. The authors also suggest that taxifolin inhibits AKT phosphorylation and thus decreases ZEB2 expression however, there is no evidence in the manuscript that links AKT phosphorylation to a decrease in ZEB2 expression. Furthermore, there is no direct link made between the expression of the miRNA and ZEB2 expression. I would suggest knockdown of the miRNA or similar style experiment to prove this.

Line 143: authors state that “wild-type beta-catenin was downregulated in the cytosol fraction and a similar trend was observed in the nuclear fraction”. Looking at the figure, this does not seem to be the case. In fact, it appears that beta-catenin increase in the nuclear fraction where the cytosol fraction decreases potentially suggesting a relocation of the protein with taxifolin.

Line 211 and 212: the authors state that there is no change in c-jun expression in the experiments however, there is no evidence in the manuscript of this. Please include a western blot.

Figure A3: The authors look at vimentin expression in response to an AKT inhibitor GSK690693. There is no evidence in the western blot to show that it in fact inhibits AKT phosphorylation at the concentrations shown. Please include pAKT expression in this blot.

Author Response

Reviewer 1

  • While quercetin did not change ZEB2 expression, the authors do not demonstrate any evidence to support this (I would recommend a western blot of ZEB2 expression with quercetin to drive home the message that only taxifolin alters ZEB2 expression) nor do they discuss a possible explanation as to why the two compounds may impose differential effects.

Response: We appreciate the reviewer recommendation as the data will show inactivity of quercetin. Therefore, we added representative western blot of ZEB2 modulation by quercetin as figure A1 into the appendix of the manuscript, lines 410-413. The discussion was supplemented for possible explanation of quercetin vs. taxifolin effects on ZEB2 expression (lines 200-201).

  • The authors also suggest that taxifolin inhibits AKT phosphorylation and thus decreases ZEB2 expression however, there is no evidence in the manuscript that links AKT phosphorylation to a decrease in ZEB2 expression.

Response: Our working hypothesis is based on two different effects of taxifolin. Firstly, taxifolin modulates the ZEB2 expression indicating the possibility of triggering EMT. Secondly, taxifolin reduces Akt activity. However, this second effect of taxifolin cannot be manifested at the level of ZEB2 expression because we would not detect any changes in the ZEB2 expression during the first experiment. The second effect of taxifolin regulates signalling downstream of ZEB2, such as vimentin, E-cadherin and N-cadherin expression. We do not see any point in showing a link between AkT phosphorylation and a decrease in ZEB2 expression.

  • .Furthermore, there is no direct link made between the expression of the miRNA and ZEB2 expression. I would suggest knockdown of the miRNA or similar style experiment to prove this.

Response: Jiang G. et al. (2017) performed experiments with miR-211 mimics and luciferase reporter fused to the wild-type 30-UTR of ZEB2. The WT plasmid activity was suppressed but mutant version of the 30-UTR was not affected. Second experiment focused on impact of miR-211-5p overexpression on ZEB2 protein levels that was downregulated. Both experiments were performed in Hep G2 cell line. This reference (Jiang et al.) is listed in Table 1 and Table 1A. Moreover, each miRNA listed in Table 1 or Table 1A has an associated reference that demonstrates its involvement in the regulation of ZEB2 protein expression. In some cases, this validation was performed in other cell lines as well.

  • Line 211 and 212: the authors state that there is no change in c-jun expression in the experiments however, there is no evidence in the manuscript of this. Please include a western blot.

Response: Representative western blot of c-jun was added into the supplementary section of the manuscript (lines 452-453).

  • Figure A3: The authors look at vimentin expression in response to an AKT inhibitor GSK690693. There is no evidence in the western blot to show that it in fact inhibits AKT phosphorylation at the concentrations shown. Please include pAKT expression in this blot.

Response: The demonstration of GSK690693 inhibitory activity by Akt phosphorylation appears to be counterproductive and rather confusing for readers. Common feature of majority of ATP-competent Akt inhibitors, such as GSK690693, is the induction of paradoxical hyperphosphorylation in both Akt regulatory sites Thr308 and Ser473, respectively. This property was observed during our experiments as well. It seems to be more reasonable to use evaluation of some downstream protein phosphorylation – GSK-3β, CREB or FoxO. Hence, we added a representative western blot demonstrating the effect of GSK690693 on the phosphorylation of CREB at position Ser133 (lines 461-463).

  • Line 143: authors state that “wild-type beta-catenin was downregulated in the cytosol fraction and a similar trend was observed in the nuclear fraction”. Looking at the figure, this does not seem to be the case. In fact, it appears that beta-catenin increase in the nuclear fraction where the cytosol fraction decreases potentially suggesting a relocation of the protein with taxifolin.

Response: Indeed, Reviewer 1 is right. There was slightly increased amount of beta-catenin in the nucleus compared to cytosol but it did not reach statistical significance.

Increased amount of beta-catenin does not necessarily mean relocation or even relocation-related activation of the TCF/Lef and beta-catenin mediated pathway. GSK-3β plays crucial role in degradation of beta catenin in the cytosol but not in the nucleus, where it forms non-functional complex with beta catenin and other proteins (Caspi M. et al. (2008)). However, GSK-3β does not cause degradation of this complex. Despite the slight relocation of beta-catenin, the effect on wnt-regulated genes may not be present.

Reviewer 2 Report

In this article, Dostal et al. described the effect of taxifolin, a polyphenol, on the expression level and biological function of ZEB2. The authors demonstrated that, although taxifolin upregulated ZEB2, presumably by regulating ZEB2-targeting miRNAs, it failed to promote ZEB2-mediated epithelial-mesenchymal transition (EMT) due to the suppression of vimentin, likely through taxifolin-induced inhibition of AKT activation. Overall, although the finding is interesting, there are several missing links needed to be established to support the conclusion. Comments on this manuscript are listed as following.

  1. Although this article started by investigating the miRNAs regulated by taxifolin treatment, consequently leading to the focus on ZEB2. However, there is no experimental evidence to validate the miRNAs upregulated or downregulated by taxifolin. Authors should present qPCR data to validate the altered levels of miRNAs following taxifolin stimulation. Furthermore, these miRNAs indeed regulate the levels of ZEB2. Lastly, whether these miRNAs are involved in taxifolin-induced upregulation of ZEB2 must be verified. This question can be addressed by either knockdown of these miRNAs or overexpression of miRNAs mimetics in the context of taxifolin-treated cells.
  2. The authors mentioned that vimentin is a downstream target of ZEB2 (Line 136). Does this mean that vimentin is a transcriptional target of ZEB2? If so, please cite the pertinent reference to support this argument.
  3. Although the authors demonstrated that taxifolin reduces the levels of phosphorylation of AKT, the authors did not reveal the residues of AKT that is phosphorylated. Also, it is recommended to examine the levels of AKT-mediated phosphorylation of GSK3β at serine 9 residue to validate the inhibition of AKT activity by taxifolin treatment.
  4. Given that multiple molecules are involved in the execution of EMT in addition to vimentin, authors should validate whether vimentin downregulation itself indeed accounts for the failure of taxifolin to induce EMT even though taxifolin upregulates ZEB2.
  5. Only one cell line, namely HepG2, was used in this study. Additional hepatocellular carcinoma cell lines should be included in this study to exclude the cell-specific effect of taxifolin regarding ZEB2 expression.

Author Response

1.Although this article started by investigating the miRNAs regulated by taxifolin treatment, consequently leading to the focus on ZEB2. However, there is no experimental evidence to validate the miRNAs upregulated or downregulated by taxifolin. Authors should present qPCR data to validate the altered levels of miRNAs following taxifolin stimulation. Furthermore, these miRNAs indeed regulate the levels of ZEB2. Lastly, whether these miRNAs are involved in taxifolin-induced upregulation of ZEB2 must be verified. This question can be addressed by either knockdown of these miRNAs or overexpression of miRNAs mimetics in the context of taxifolin-treated cells.

Response: All miRNA mentioned in Table 1 and Table 1A were experimentally validated by different researchers and were published in many articles. These articles are cited in reference part of both tables. These experiments were done with Hep G2 or other cell lines. Some of these miRNAs have a citation for the MiRTarBase database that provides information on the validated targets of individual miRNAs, including the publication and type of validation method (only strong methods such as reporter assays, western blots or qPCR were used).

Moreover, miRNA array part of the study was only the minor part. The major part of the study was to evaluate ZEB2 signaling. Legends of both tables newly contain a text which emphasizes the individual references as the source where the ZEB2/miRNA validation was performed (lines 101-102 and 407-408).

    We considered validation of all miRNAs with significantly altered expression by qPCR as unnecessary. During our experiments, we performed three independent measurements for each substance (one substance = three chips), which is an above-standard. Authors should present qPCR data to validate the altered levels of miRNAs following taxifolin stimulation. Furthermore, these miRNAs indeed regulate the levels of ZEB2. Lastly, whether these miRNAs are involved in taxifolin-induced upregulation of ZEB2 must be verified. This question can be addressed by either knockdown of these miRNAs or overexpression of miRNAs mimetics in the context of taxifolin-treated cells. We decided to use miRNA precursors pre-treatment in combination with taxifolin incubation. The results of these experiments showed correlation between taxifolin and miR-377 precursors transfection and are described in separate paragraph of results section (lines 159-173). Finally, the discussion was extended with the necessary text as well (lines 245-253).

2.The authors mentioned that vimentin is a downstream target of ZEB2 (Line 136). Does this mean that vimentin is a transcriptional target of ZEB2? If so, please cite the pertinent reference to support this argument.

Response: In view of this comment we cited the relevant reference (Lines 139 and 209) showing that ZEB2/SIP1 cDNA transfection modulates activity of vimentin reporter plasmid.

3. Although the authors demonstrated that taxifolin reduces the levels of phosphorylation of AKT, the authors did not reveal the residues of AKT that is phosphorylated. Also, it is recommended to examine the levels of AKT-mediated phosphorylation of GSK3β at serine 9 residue to validate the inhibition of AKT activity by taxifolin treatment.

Response: The position of the phosphorylated Akt residue that was observed during the experiment was added into the legend of Figure 4 (line 154) and Materials and methods section (line 361). Position of phosphorylated residue of Akt was Ser473. The demonstration of GSK690693 inhibitory activity by Akt phosphorylation appears to be counterproductive and rather confusing for readers. Common feature of majority ATP-competent Akt inhibitors, such as GSK690693, is the induction of paradoxical hyperphosphorylation in both Akt regulatory sites Thr308 and Ser473 respectively. It seems to be more reasonable to use evaluation of some downstream protein phosphorylation. Since GSK-3β is unavailable to us at this time, we chose phosphorylation of another downstream target CREB. Hence, we added a representative western blot to the manuscript demonstrating the effect of GSK690693 on the phosphorylation of CREB (lines 461-463).

4.Given that multiple molecules are involved in the execution of EMT in addition to vimentin, authors should validate whether vimentin downregulation itself indeed accounts for the failure of taxifolin to induce EMT even though taxifolin upregulates ZEB2.

Response: Indeed, multiple molecules are involved in the execution of EMT, as we described in the manuscript. We agree with the reviewer's remark and we improved our discussion (lines 241-244) about Akt crosstalk with other proteins involved in EMT. Reference demonstrates that reduced Akt activity results in upregulation of E-cadherin and downregulation of N-cadherin indicating the complex impact of the Akt activity on EMT.

5. Only one cell line, namely HepG2, was used in this study. Additional hepatocellular carcinoma cell lines should be included in this study to exclude the cell-specific effect of taxifolin regarding ZEB2 expression.

Response: We thank the reviewer for pointing this out as we had the very original title more focused. Based on the comment we returned to our original title „Dual effect of taxifolin on ZEB2 cancer signaling in Hep G2 cells”.

Reviewer 3 Report

The authors evaluated the effect of taxifolin and quercetin on ZEB2 cancer signaling, determining the expression of miRNA linked to the modulation of ZEB2 expression. The experiment is well designed and results are interesting. Conclusions are supported by the results obtained. My only concern is about the inclusion of quercetin in the experiment, and the lack of discussion of the results obtained for quercetin. Also, there is no comparison between the results obtained for taxifolin and quercetin, and why results were different. Authors should discuss based on the differences on the chemical structure of both polyphenols. Also, it is important to point out in the manuscript that the chemical structure of the compounds would be modified due to digestion processes, quercetin and taxifolin are bioavailable? can these compounds reach the target organ? do they suffer chemical modifications? the results obtained in this in vitro study could be replicated in vivo?. 

Author Response

Quercetin data were included in the manuscript as well as discussion that gives a possible explanation for the observed difference in ZEB2 protein expression between quercetin and taxifolin. See answers to reviewer 1, paragraph 
    Authors should discuss based on the differences on the chemical structure of both polyphenols. Also, it is important to point out in the manuscript that the chemical structure of the compounds would be modified due to digestion processes, quercetin and taxifolin are bioavailable? can these compounds reach the target organ? do they suffer chemical modifications? the results obtained in this in vitro study could be replicated in vivo?.

Our study was focused exclusively on in vitro experiments therefore we did not venture into discussing bioavailability, metabolism, and target organs. The use of human hepatocytes in our case served only as a non-proliferating primary cell culture for comparison with HepG2 cells. We did not consider it necessary to include the discussion in order to maintain the focus on in vitro effect.

Round 2

Reviewer 1 Report

I thank the authors for addressing the comments and making the relevant amendments to the manuscript. 

Author Response

We thank the Reviewer for accepting our detailed response and amendments to the manuscript.

Reviewer 2 Report

The Reviewer thank the authors for responding to my comments on the original manuscript. Comments on this revised manuscript are listed below.

Major:

Figure 5 was intended to validate the role of miR-377 downregulation by taxifolin accounts for taxifolin-induced upregulation of ZEB2. Logically, ectopic expression of miR-377 is expected to counteract the promoting effect of taxifolin on ZEB2 expression. However, the results shown in Figure 5 was that miR-377 expression barely abolishes taxifolin-elicited increase of ZEB2 levels, a result that excludes the role of miR-377 reduction in ZEB2  upregulation upon taxifolin treatment and therefore contradicts to the conclusion of this study.

Minor:

For clarity to the readers, in the Abstract section, it is highly recommended to list the four miRNAs downregulated by taxifolin in HepG2 cells.

Author Response

Response to the Major comment:

The new Figure 5 shows the expected effect of taxifolin. We show that miR-377 down-regulates ZEB2. The effect, which in your comment is described as "barely abolishes", is statistically significant and reproducible. We would not expect more pronounced effect as miRNAs have multiple targets, and contrary to siRNA, are not directed exclusively towards one target mRNA. Moreover, the target mRNA is not necessarily destroyed, stored in p-bodies and potentially released and transcribed, although at much lower frequency. Hence the approximately 40% down-regulation is reached. Then we go on to show that taxifolin counteracts this effect by restoring the original level of expression of ZEB2 even in the presence of miR-377. We feel that i does support our conclusion. Taxifolin down-regulates miR-377 thereby allowing increased ZEB2 expression. 

Response to the Minor comment:

Thank you for pointing that out. It will be helpful to the Readers and we will list all four miRNAs in the Abstract.